# Scaling Language Model Size in Cross-Device Federated Learning

**Jae Hun Ro**[*] and **Theresa Breiner** and **Lara McConnaughey**
**Mingqing Chen** and **Ananda Theertha Suresh** and **Shankar Kumar** and **Rajiv Mathews**

**Google**

*jaero@google.com

## Abstract

Most studies in cross-device federated learning focus on small models, due to the server-client communication and on-device computation bottlenecks. In this work, we leverage various techniques for mitigating these bottlenecks to train larger language models in cross-device federated learning. With systematic applications of partial model training, quantization, efficient transfer learning, and communication-efficient optimizers, we are able to train a 21M parameter Transformer that achieves the same perplexity as that of a similarly sized LSTM with $\sim 10\times$ smaller client-to-server communication cost and $11\%$ lower perplexity than smaller LSTMs commonly studied in literature.

## 1 Introduction

Federated learning is a distributed training technique, where a model is trained on data distributed across clients or edge devices without user-generated data ever leaving the device, providing an additional layer of privacy and security (Konečný et al., 2016b,a; McMahan et al., 2017). We refer readers to (Li et al., 2020; Kairouz et al., 2021) for a detailed literature survey on federated learning. Federated learning has been used in several applications including virtual keyboard applications (Hard et al., 2018), keyword spotting (Hard et al., 2020), and healthcare (Brisimi et al., 2018).

Language models (LM) have many uses in language-based applications including virtual keyboard (Chen et al., 2019; Zhang et al., 2021) and automatic speech recognition (Kannan et al., 2018; Variani et al., 2020; Gruenstein et al., 2021). Recently, there has been increased interest in training progressively larger and deeper LMs with impressive quality improvements in downstream tasks, including question answering, text classification, and text summarization (Devlin et al., 2019; Dai et al., 2019; Yang et al., 2019; Irie et al., 2019; Ka-

plan et al., 2020). These models tend to be variants of the Transformer (Vaswani et al., 2017).

Federated learning is typically studied in two scenarios: *cross-silo*, where the number of clients is small, and *cross-device*, where the number of clients can be in the order of millions (Hard et al., 2018). In this work we focus on cross-device, where devices are typically edge devices such as cell phones, with limited computation and communication capabilities. Hence, the major benchmark LMs tend to be very limited in size (McMahan et al., 2017, 2018; Caldas et al., 2019a; Reddi et al., 2020; Sim et al., 2021) because memory, computation, and communication are critical bottlenecks (Kairouz et al., 2021). In particular, previous works that train federated LMs in production settings have used coupled input forget gate (CIFG) long short-term memory (LSTM) models with fewer than 4 million parameters (Hard et al., 2018; Chen et al., 2019; Ramaswamy et al., 2020). These resource constraints have motivated research into various efficient algorithms for training larger models with federated learning (Konečný et al., 2016b; Hamer et al., 2020). However, most of these techniques are still evaluated on relatively small models compared to their server-based counterparts. In this work, we systematically evaluate multiple strategies for mitigating communication and computation costs of training larger LMs to determine if the impressive quality gains from larger models can also be achieved in cross-device federated learning.

While there are previous works on *efficient* Transformers (Tay et al., 2020, 2021), we forgo these efficient variants as they may actually be more inefficient when sequences are short (Katharopoulos et al., 2020; Choromanski et al., 2021). Additionally, Lin et al. (2020); Liu and Miller (2020); Hilmkil et al. (2021) trained large Transformer models in the cross-silo setting, where devices have more resources, whereas we focus on the resource-constrained cross-device setting.

Recent large LMs, such as GPT-3 (Brown et al., 2020), contain hundreds of billions of parameters, which is substantially bigger than the memory limits of edge devices. Therefore in this work, we consider *large* models to be at most 25 million parameters, which is still considerably larger than existing models trained on-device.

The rest of the paper is organized as follows. In Section 2, we overview our contributions. In Section 3, we detail the dataset and models. We then analyze techniques to reduce the per-round cost in Section 4, and the number of communication rounds in Section 5. Finally in Section 6, we combine techniques and demonstrate that large Transformers can be trained using many fewer rounds and significantly lower communication and computation cost.

## 2 Our contributions

We explore two regimes: small models typically studied in cross-device federated learning with fewer than 5M parameters and new larger models with at most 25M parameters. We study two architectures: CIFG-LSTM (Hochreiter and Schmidhuber, 1997), or LSTM for simplicity, (Hard et al., 2018) and Transformer (Vaswani et al., 2017). Our contributions are the following:

- We are the first to investigate Transformer LMs with 25M parameters for cross-device federated learning, which we find outperform LSTMs of similar size.

- We demonstrate that large models substantially outperform small models on standard tasks but at much higher communication and computation costs, requiring $4\times$ the communication cost per round.

- We investigate quantization and partial model training to address the per round communication and computation cost. With quantization, we achieve similar perplexity with half the download cost and one quarter of the upload cost, reducing total communication cost by $62.5\%$. Partial model training can further reduce the upload cost by $60\%$.

- We study transfer learning as a method of reducing the number of communication rounds and show that centralized pretraining on a suitable alternate corpus reduces the total communication rounds by $3\times$.

- We show that the combination of above techniques can be used to train a Large Transformer with the same perplexity as that of a similarly sized LSTM with $\sim 10\times$ the smaller client-to-server communication cost.

## 3 Dataset and models

In this section, we describe the models and dataset used in the rest of the paper. We train on the Stack Overflow federated dataset from TFF (2018), which contains posts from the public forum grouped by username. Following trends in training Transformers, we use sentence-piece (Kudo and Richardson, 2018) for sub-word tokenization with a vocabulary size of 4K. The sentence-piece model is computed based on the entire Stack Overflow training corpus in an offline process on server. During federated learning, this fixed sentence-piece model is transmitted to each client to encode the local text data. Doing so provides greater coverage for cross-dataset applications as well as potential downstream speech applications such as ASR (Li et al., 2021; Sim et al., 2021). We measure performance on next-subword prediction using test perplexity. See Appendix A for descriptive dataset statistics. All experiments were implemented using JAX (Bradbury et al., 2018) and FedJAX (Ro et al., 2021) federated simulation libraries.

We first did a hyperparameter search for each model and size ($\leq$ 5M and $\leq$ 25M), with FedAdam (Reddi et al., 2020), or FedAvg for simplicity, with 200 clients per round for 3K rounds, resulting in four models: *Small LSTM* (4.7M), *Large LSTM* (18.8M), *Small Transformer* (4.1M), and *Large Transformer* (21M).

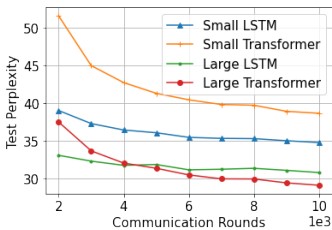

Figure 1: Test perplexity over communication rounds for each class and size of model.

We then trained the chosen architectures with 800 clients per round for 10K rounds in Figure 1. As expected, the larger variants significantly outperform their smaller counterparts with the Large Transformer achieving the best perplexity. However, the larger models are more expensive to train

per round and although the Large Transformer achieves the best perplexity, it only surpasses the Large LSTM after 4K rounds. Next, we focus on techniques to reduce this cost per round and number of rounds. For more details about the architecture search, the selected models, and their performance, see Appendix A.

# 4 Cost per round

The larger models have 18.8M and 21M parameters (150MB and 168MB, at 32 bits per parameter) which need to be downloaded, trained, and uploaded at each round, a strain on both communication and computation on device. There are often strict time or transfer byte limits for each round of training, which can prohibit some devices from training these models due to slower transfer/processing speeds (Kairouz et al., 2021). We show that we can significantly reduce these costs by partial model training and quantization techniques.

**Partial model training**: Training only a subset of the model can reduce the computational cost of training and has been examined in both federated (Caldas et al., 2019b; Yang et al., 2021) and non-federated (Kovaleva et al., 2019) settings. Additionally, reducing the number of trainable parameters can also decrease communication cost since only the trainable parameters need to be uploaded.

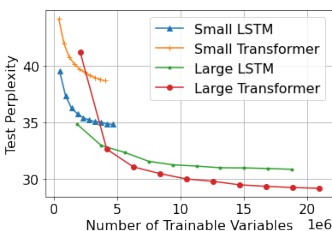

Figure 2: Test perplexity as a function of number of trainable variables.

We follow the Partial Variable Training (PVT) per client per round strategy (Yang et al., 2021) as it only freezes a subset of the original model and can be applied generally to multiple model architecture types. For more experiment details, see Appendix B. We report test perplexity as a function of number of trainable variables in Figure 2. Large LSTM seems to be able to handle more aggressive parameter freezing compared to Large Transformer in terms of quality regression. However, training only 40% of variables for the Large Transformer (6.3M) achieves better performance than the full Large LSTM (18.8M).

**Quantization**: To reduce communication costs, various quantization strategies can decrease the number of bits required to represent model parameters (Bernstein et al., 2018; Reisizadeh et al., 2020; Gandikota et al., 2021; Vargaftik et al., 2021). We examine stochastic k-level uniform quantization (Alistarh et al., 2017; Suresh et al., 2017) as it can be applied to model parameters on download (server-to-client) and model updates on upload (client-to-server) communication with adjustable levels of compression, and compare with TernGrad, an upload technique (Wen et al., 2017).

We focus analysis on larger models which are more affected by quantization. The LSTM appears more "quantizable" during download than the Transformer, with less regression in Figure 3. The perplexity of the Transformer with 16 download bits matches that of the baseline Transformer and with 12 bits its perplexity is close to that of the LSTM.

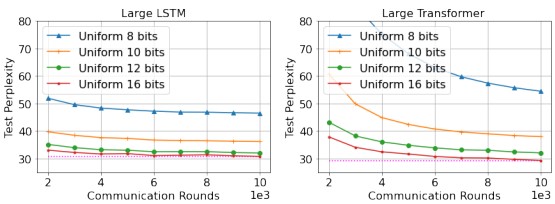

Figure 3: Test perplexity over communication rounds for varying download quantization levels, with upload quantization fixed to 8 bits. Dashed line shows the baseline without quantization.

For both the models, 8 bit upload matches the corresponding baselines, or even 6 bits for the LSTM in Figure 4. TernGrad, requiring $\log_2(3)$ bits, outperforms the 4 bit in the Transformer but not for the LSTM in Figure 5. More details are in Appendix C.

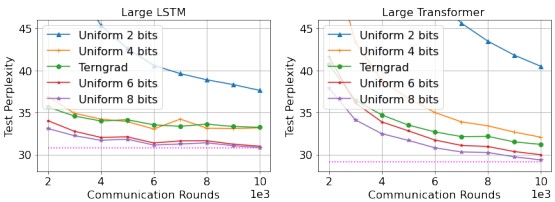

Figure 4: Test perplexity over communication rounds for varying upload quantization levels, with download quantization fixed to 16 bits. TernGrad is comparable to uniform with about 1.6 bits. Dashed line shows the baseline without quantization.

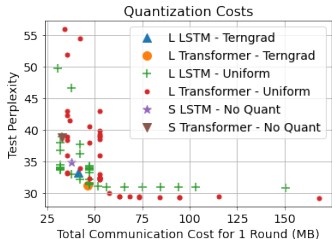

Figure 5: Test set perplexity versus total communication cost (download + upload) in a single round of training, for each quantization algorithm. Uniform settings include points for varying quantization bits.

# 5 Number of communication rounds

**Transfer learning**: Transfer learning leverages pretrained models to improve model quality (Houlsby et al., 2019). By pretraining, the number of communication rounds required for model convergence can be significantly reduced (Stremmel and Singh, 2020).

We use two datasets for pretraining: a large corpus of digitized books (Zhang et al., 2021) and the One Billion Word Benchmark (LM1B) (Chelba et al., 2014). After pretraining using synchronous SGD for 30M steps, we finetune on Stack Overflow using FedAvg. For additional details, see Appendix D. We report results for each of the pretraining datasets and random initialization in Figure 6.

Books consistently outperforms LM1B for both the LSTM and Transformer. Pretraining greatly benefits the Large Transformer compared to the Large LSTM, reducing the number of rounds needed to reach the final 10K without pretraining by 4K rounds. Furthermore, at round 2K, the Large Transformer already outperforms the Large LSTM, making the number of rounds needed for training similar to that of smaller models used in mobile keyboard prediction (Hard et al., 2018).

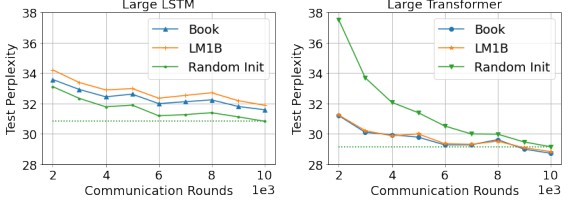

Figure 6: Test perplexity over communication comparing pretraining corpora. Dashed line is the final perplexity reached by the randomly initialized model.

**Different optimizers**: Since the introduction of FedAvg, several variations continue to be developed (Li et al., 2018; Hamer et al., 2020; Reddi

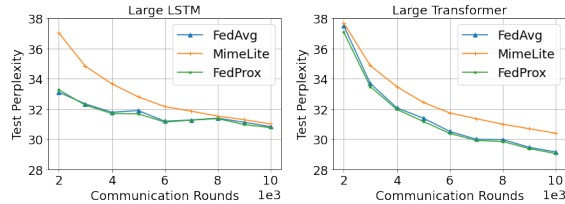

Figure 7: Test perplexity over communication rounds for each model and algorithm.

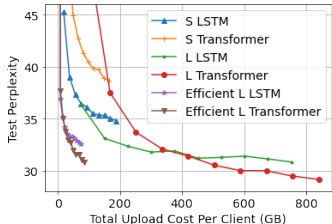

Figure 8: Test perplexity over total uploaded gigabytes per client for each class of model.

et al., 2020). Specifically, we examine MimeLite (Karimireddy et al., 2020) and FedProx (Li et al., 2018) as they have been shown to reduce the total amount of rounds required for provable convergence. However, in Figure 7, FedProx and MimeLite do not improve convergence speed over FedAvg. More details can be found in Appendix E.

# 6 Combination of techniques

We experiment with combining partial model training, quantization, and transfer learning to train *efficient* larger models. For these experiments, we train on just 40% of trainable parameters with PVT and warm start after pretraining on the Books corpus. Combining download quantization with these techniques did not perform as well, so we only apply 8 bit uniform quantization on upload, which is the tightest communication bottleneck (Statista.com (2021) reports that mobile upload speeds worldwide are over 4× slower than download as of May 2021). For the full experiment details, refer to Appendix F. We report the test perplexity in terms of total upload communication cost in Figure 8. Restricting for small upload costs (< 200GB), the efficient models outperform all others with the efficient Large Transformer yielding the best perplexity. Furthermore, the efficient Large Transformer also achieves the same perplexity as the Large LSTM with no efficient techniques.

## 7 Conclusion

We systematically studied several techniques for addressing the communication and computation bottlenecks of federated learning. We further demonstrated that these techniques, individually or in combination, can scale to larger models in cross-device federated learning. Extending this study to other architectures and efficient strategies remains an interesting open question.

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

# Appendix

## A  Dataset and models

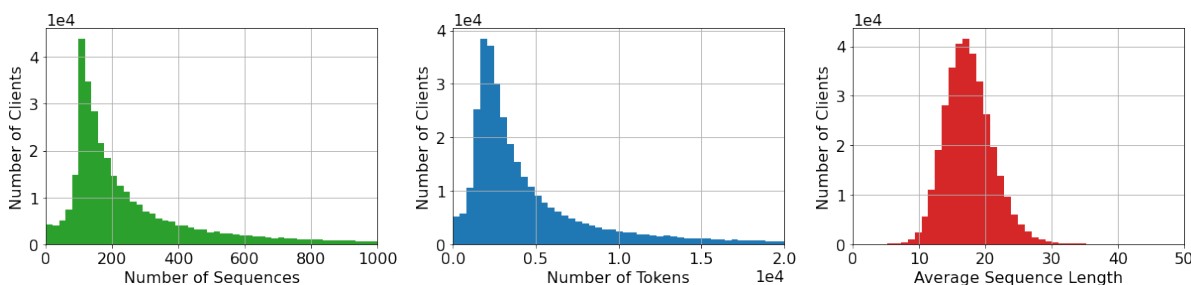

Figure 9: Stack Overflow train split sub-word statistics.

Table 1: Selected architectures for each model and size range. The values in [ ] are the possible hyperparameter values searched over. Layer Size refers to the LSTM layer dimension and MLP layer dimension for Transformer and # Layers refers to number of LSTM layers and number of Transformer blocks.

| Model | # Parameters | Embedding Size $[128, 256, 512, 1024]$ | Layer Size $[512, 1024, 2048]$ | # Layers $[1, 2, 4, 6, 8]$ |
|---|---|---|---|---|
| Small LSTM | 4.7M | 256 | 2048 | 1 |
| Small Transformer | 4.1M | 128 | 2048 | 6 |
| Large LSTM | 18.8M | 1024 | 2048 | 1 |
| Large Transformer | 21.0M | 512 | 2048 | 6 |

Table 2: Test metrics after 10K rounds of training for each class of model and number of clients per round. The results in **bold** indicate the best for each size range.

| Model | # Clients | Perplexity |
|---|---|---|
| Small LSTM | 200 | 35.31 |
| Small LSTM | 400 | 34.93 |
| Small LSTM | 800 | **34.80** |
| Small Transformer | 200 | 40.18 |
| Small Transformer | 400 | 39.38 |
| Small Transformer | 800 | 38.66 |
| Large LSTM | 200 | 30.97 |
| Large LSTM | 400 | 30.79 |
| Large LSTM | 800 | 30.83 |
| Large Transformer | 200 | 30.64 |
| Large Transformer | 400 | 29.81 |
| Large Transformer | 800 | **29.15** |

For the baseline architecture search, Table 1 details the selected architectures as well as the search ranges for each dimension. The final hyperparameters were selected based on the test perplexity after 3K rounds of training using FedAvg with 200 clients per round. From here on, we fix the Adam optimizer with $\beta_1$ at 0.9, $\beta_2$ at 0.999, and epsilon at $1e^{-8}$. Additionally, based on the distribution of average sequence lengths across Stack Overflow clients in Figure 9, we fix the max sequence length for training and evaluation to 30.

Table 2 contains the results for each selected model after 10K rounds of training using FedAvg with 200, 400, and 800 clients per round. As expected, the best results are achieved by using 800 clients per round. Thus, from here on, we report results for 800 clients per round only. For these experiments, we

Table 3: Selected hyperparameters for each model and size range. The values in [ ] are the possible hyperparameter values searched over. Batch Size, # Examples, and Clipnorm here apply to the client local SGD steps. LR is learning rate.

| Model | Batch Size [8, 16] | # Examples [1200, 1600] | Clipnorm [0.0, 16.0] | Client LR [0.01, 0.1, 0.5, 1.0, 2.0] | Server LR [0.001, 0.01] |
|---|---|---|---|---|---|
| Small LSTM | 16 | 1200 | 16.0 | 1.0 | 0.001 |
| Small Transformer | 16 | 1200 | 0.0 | 0.1 | 0.001 |
| Large LSTM | 16 | 1200 | 16.0 | 1.0 | 0.001 |
| Large Transformer | 16 | 1200 | 0.0 | 0.5 | 0.001 |

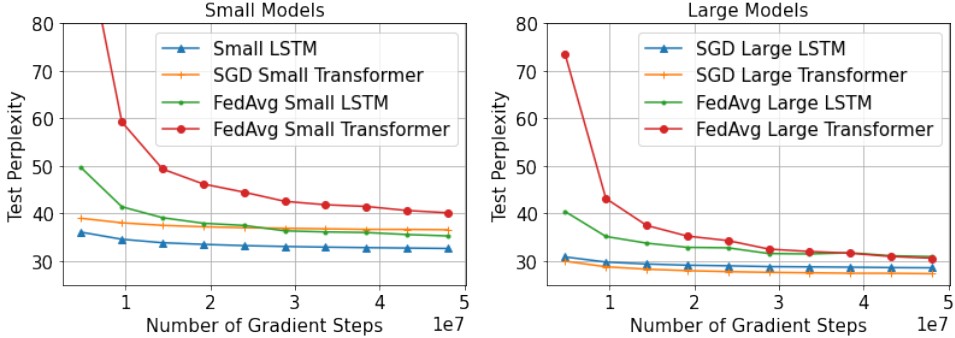

Figure 10: Test set perplexity as a function of number of gradient computations for comparing the centralized and federated averaging baselines.

also search over client learning rate, client batch size, client max number of examples (with client number of epochs fixed to 1), client $\ell_2$ norm for clipping, and server learning rate. The search ranges as well as selected values for each model are detailed in Table 3. For all following experiments, we fix client batch size to 16 and client max number of examples to 1200 since the larger batch size consistently performed the best and Figure 9 shows that 1200 sequences is more than enough to cover the vast majority of clients with the number of epochs fixed at 1. We also search over the same ranges for all following experiments where applicable for consistency.

As an additional baseline comparison, we also train each model using synchronous SGD to observe model quality in terms of number of gradient computations. These centralized baselines provide a rough estimate of an upper bound on model quality for federated learning. To produce a reasonable comparison between the federated and centralized experiments, we compare by number of gradient computations. We approximate the number of gradient steps taken for federated learning with 200 clients per round for 10K communication rounds. We train the centralized models using the Adam optimizer and run periodic evaluation on the test set at the same frequency as the federated experiments. We report and compare final metrics between centralized training and federated averaging on the test set in Figure 10. Observing the test perplexity over gradient steps, it is evident that the relative rankings of the models remain consistent between centralized and federated baselines. Additionally, by 10K rounds, the large federated models seem to approach somewhat close in perplexity to their centralized counterparts.

## B Partial model training

In our experiments with PVT, we vary the percentage of trainable variables from 10% to 90% in increments of 10. As before, we search over the hyperparameters in Table 3 and find them to be mostly consistent with baseline other than client learning rate. Following Yang et al. (2021), we use the per client per round (PCPR) configuration, where the frozen variables vary from round to round and from client to client, as this was shown to achieve the highest accuracy. Specifically, we only freeze subsets of the multiplicative vectors and matrices of the original model. This corresponds to the embedding and weights of the LSTM, and for the Transformer, the weights of the MLP layer, attention matrices, layer normalization in each

Table 4: Test perplexity after 10K communication rounds of training for each class of model and PVT % of trainable variables.

| Model | Trainable % | # Parameters | Perplexity |
|---|---|---|---|
| Small LSTM | 100% | 4.7M | 34.80 |
| Small Transformer | 100% | 4.1M | 38.66 |
| Large LSTM | 100% | 18.8M | 30.83 |
| Large LSTM | 40% | 7.5M | 31.53 |
| Large LSTM | 20% | 3.8M | 32.93 |
| Large Transformer | 100% | 21.0M | 29.15 |
| Large Transformer | 40% | 8.4M | 30.45 |
| Large Transformer | 20% | 4.2M | 32.61 |

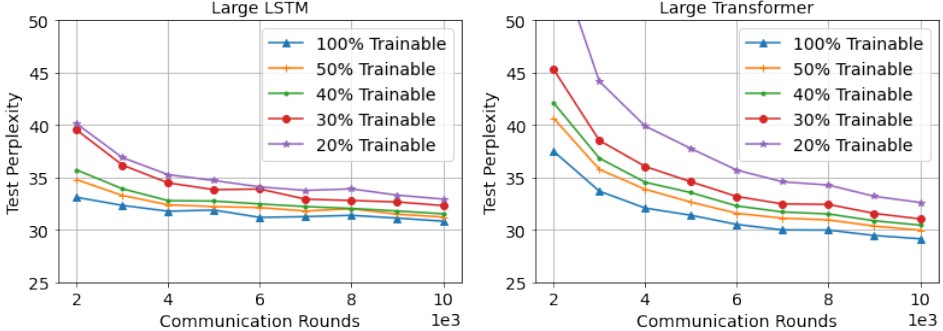

Figure 11: Test perplexity over communication rounds for the large models with select percentages of trainable variables denoted by $X\%$ with $100\%$ indicating all trainable variables are trained (i.e. baseline).

block, and embedding. We also note though that although overall the number of trainable variables might average to the desired percentage (e.g. $10\%$), for certain architectures, like LSTM, that don't have that many *freezable variables* (only one layer's weight matrix and embedding matrix), the number of trained variables will be much more variable from round to round. On the other hand, for architectures, like Transformer, that have more freezable variables (6 blocks' weight matrices and attention matrices and embeddings), the number of trained is much more consistent between rounds.

We report test set perplexity over communication rounds for the large architectures and varying degrees of PVT in Figure 11 with the number of clients per round set to 800. Looking at Table 4, it is evident that both large models can handle some percentage of partial freezing up until a certain point and that the Large Transformer with only $40\%$ of trainable variables can reach a similar perplexity as the Large LSTM with $100\%$ trainable variables by 10K rounds or so. However, training for the full 10K rounds can be a communication bottleneck so PVT would need to be combined with another technique to reduce the number of rounds needed.

## C   Quantization

In stochastic $k$-level uniform quantization (Suresh et al., 2017), values in each layer are converted into one of $k$ evenly distributed values between the layer min and max, stochastically assigned to the closest target value either above or below the real value. The lower the $k$ value, the more the data is being compressed, as the number of bits used to store the value equals $\log_2(k)$. For download quantization, we explore $k$ values corresponding to between 8 and 28 bits. For upload quantization, which can be a larger bottleneck in edge devices (Statista.com, 2021), we explore $k$ values corresponding to between 1 and 28 bits. On upload, we also try applying zero-centering during uniform quantization as well as trying the TernGrad (Wen et al., 2017) algorithm, which quantizes values in each vector $v$ into only one of three values, 0 and $\pm \max(|v|)$, corresponding to $\log_2(3)$ ($\sim 1.585$) bits per parameter. While TernGrad is designed to use L infinity clipping ($\ell_\infty$), we experiment with and without this for completeness.

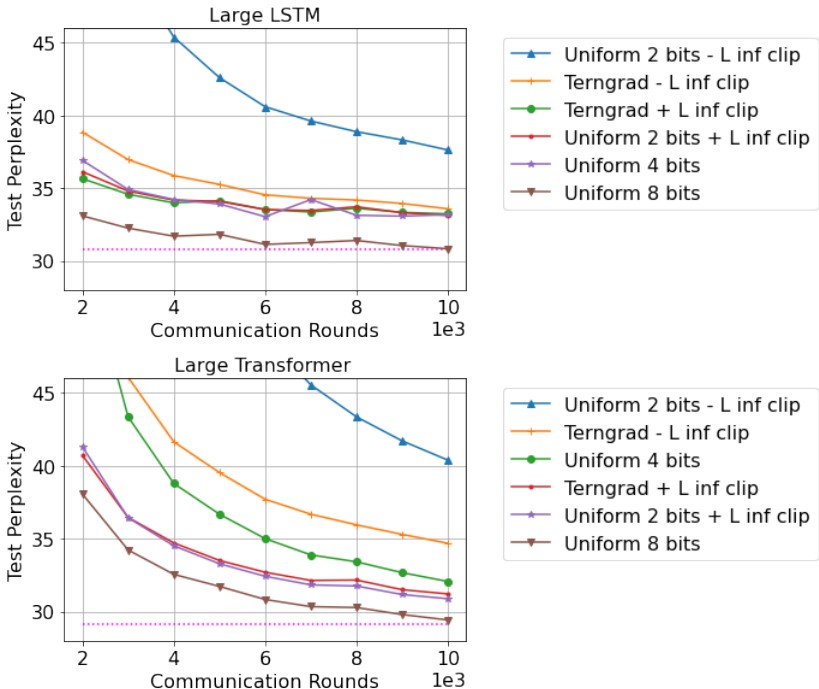

Figure 12: Test set perplexity over communication rounds for varying upload quantization levels, with download quantization fixed to 16 bits. The dotted line shows baseline perplexity achieved after 10K rounds without any quantization.

While $\ell_\infty$ clipping did make a significant difference in the TernGrad experiment for Transformers, performing much better with it than without, it did not have a large effect on the TernGrad performance in the LSTM in Figure 12. TernGrad and its counterpart uniform quantization to $\sim 1.585$ bits performed the same, as long as $\ell_\infty$ clipping was applied. It is clear from the uniform 2-bit experiments as well that $\ell_\infty$ clipping is important when quantizing into these lower number of bits; the 2-bit experiment without clipping performs much worse than the Terngrad without clipping, although enabling clipping allows 2-bit to perform slightly better than Terngrad's $\log_2(3)$ bits with clipping. Zero-centering did not seem to affect upload behavior much for either model, marginally improving the LSTM and marginally degrading the Transformer.

We explore the patterns of communication cost for each experiment setting in Figure 5. We calculate the approximate download and upload MB for each experiment by multiplying the model's number of parameters by the number of download or upload bits to get total bits transported.

Examining Figure 5, we note the baseline points for each set of experiments as the lowest and rightmost, getting the best perplexity but also highest communication cost. Starting from there, we see trends of no perplexity degradation as we apply conservative quantization to the Large LSTM and Transformer settings and move left in the plot. We then reach an elbow in the points for each setting right around where the Terngrad point is, from which point perplexity degrades drastically without much communication cost savings as the points head up in two lines as upload quantization is reduced, with one line corresponding to experiments with download 16 bits and the other to download 12 bits. While the Terngrad point for the Large Transformer falls at the outermost point in the "elbow" and therefore gives the best tradeoff for cost versus perplexity, there is one uniform quantization point that does better than the Large LSTM Terngrad, which is download 12 bits and upload 6 bits. It makes sense that this does well as we saw that the LSTM was able to use these settings without much regression from the baseline performance, while the Transformer could only quantize to 16 download bits and 8 upload bits without regressions.

Table 5: Selected hyperparameters for each centrally trained model and dataset. The values in [ ] are the possible hyperparameter values searched over.

| Model | Dataset | Clipnorm [0, 16] | Learning Rate $[1e^{-5}, 5e^{-5}, 1e^{-4},$ $5e^{-4}, 1e^{-3}, 5e^{-3}, 1e^{-2}]$ |
|---|---|---|---|
| Small LSTM | Book | 16.0 | $5e^{-5}$ |
| Small LSTM | LM1B | 0.0 | $5e^{-5}$ |
| Large LSTM | Book | 0.0 | $5e^{-5}$ |
| Large LSTM | LM1B | 0.0 | $5e^{-5}$ |
| Small Transformer | Book | 0.0 | $1e^{-4}$ |
| Small Transformer | LM1B | 16.0 | $1e^{-4}$ |
| Large Transformer | Book | 16.0 | $5e^{-5}$ |
| Large Transformer | LM1B | 16.0 | $5e^{-5}$ |

## D  Transfer learning

To find the best models pretrained on the Books and LM1B datasets, we train for 30M steps of synchronous SGD searching over learning rate and clip norm. Like our other centrally trained models, the batch size is fixed to 16 and Adam is used with $\beta_1$ at 0.9, $\beta_2$ at 0.999, and epsilon at $1e^{-8}$. See Table 5 for the selected hyperparameters.

Next we warmstart each models with the parameters from the best corresponding pretrained centralized model and train using FedAvg for 10K rounds. We sweep over clip norm and client learning rate. See Table 6 for the selected hyperparameters. Clip norm is omitted in Table 6, since for all hyperparameter sweeps 16 was the best value. The Book dataset outperforms the LM1B dataset in all model architectures across LSTM and Transformer. Investigating the difference between the two datasets and their similarities to the Stackoverflow dataset to determine why Books always outperformed LM1B remains an interesting open question.

## E  Different optimizers

In an effort to improve communication efficiency of the larger language models, we examine two communication-efficient federated algorithms: MimeLite and FedProx. By comparing the speed and point of convergence of these algorithms in number of rounds, we can determine if the overall communication cost of training can be decreased. As before, we fix the model architectures for each class of model and conduct a basic search over learning hyperparameters using the same common search space as Table 3 with the addition of the following algorithm specific hyperparameter sweeps. For MimeLite, we use Adagrad (Duchi et al., 2011) for the base optimizer as this setup was shown to perform the best by Karimireddy et al. (2020) for Stack Overflow. For the MimeLite Adagrad base optimizer, we sweep over base learning rates of $[0.01, 0.03, 0.1, 0.3, 1.0]$ and epsilons of $[1e^{-1}, 1e^{-3}, 1e^{-5}, 1e^{-7}]$ and fix the server learning rate to 1.0. For FedProx, we sweep over $\mu$ values of $[0, 0.1, 0.01, 0.001, 0.0001]$ which controls the weight of the L2 squared norm.

We report test perplexity over 10K federated training rounds with 800 clients per round in Figure 7 and Table 7. While FedProx does slightly outperform FedAvg, it does not significantly alter the speed of training in terms of number of communication rounds. Thus, we chose to continue using FedAvg in the combination experiments for consistency across experiments and more accurate comparisons.

## F  Combination of techniques

For the combination experiments, we conducted a joint search over a smaller range of hyperparameters for each technique to keep the total search space reasonable. For PVT, we restricted the possible percentages to 20%, 30%, and 40% of trainable variables as those were shown to yield good performance while cutting model size to less than half the original size. For uniform quantization, we restricted the search of

Table 6: Test set metrics after 10K communication rounds of training for each class of model and pretrain dataset. The client learning rate listed is the best performing learning rate found from a hyperparameter sweep. Reported Δ metrics are the change in quality relative to Table 2.

| Model | Dataset | # Clients | Client Learning Rate [0.01, 0.1, 0.5, 1.0, 2.0] | Δ Perplexity |
|---|---|---|---|---|
| Small LSTM | Book | 200 | 1.0 | 0.24 |
| Small LSTM | Book | 400 | 0.5 | 1.09 |
| Small LSTM | Book | 800 | 0.5 | 1.66 |
| Small LSTM | LM1B | 200 | 1.0 | 0.53 |
| Small LSTM | LM1B | 400 | 0.5 | 1.72 |
| Small LSTM | LM1B | 800 | 0.5 | 2.36 |
| Large LSTM | Book | 200 | 0.5 | 0.59 |
| Large LSTM | Book | 400 | 0.1 | 0.79 |
| Large LSTM | Book | 800 | 0.5 | 0.94 |
| Large LSTM | LM1B | 200 | 0.5 | 0.91 |
| Large LSTM | LM1B | 400 | 0.1 | 1.09 |
| Large LSTM | LM1B | 800 | 0.5 | 1.3 |
| Small Transformer | Book | 200 | 0.1 | 0.35 |
| Small Transformer | Book | 400 | 0.1 | 1.83 |
| Small Transformer | Book | 800 | 0.1 | 3.34 |
| Small Transformer | LM1B | 200 | 0.1 | 0.42 |
| Small Transformer | LM1B | 400 | 0.1 | 1.97 |
| Small Transformer | LM1B | 800 | 0.1 | 3.49 |
| Large Transformer | Book | 200 | 0.5 | **−1.92** |
| Large Transformer | Book | 400 | 0.1 | **−0.76** |
| Large Transformer | Book | 800 | 0.1 | **−0.04** |
| Large Transformer | LM1B | 200 | 0.1 | **−1.81** |
| Large Transformer | LM1B | 400 | 0.1 | **−0.64** |
| Large Transformer | LM1B | 800 | 0.1 | 0.14 |

upload to 6 or 8 bits and download to 16 or 32 bits since the Transformer was shown to be able to handle aggressive upload quantization but required more care on download quantization. Finally, for transfer learning, we warmstarted after pretraining on the Books corpus. As in previous experiments, we also search over the common hyperparameter space defined in Table 3, where applicable.

Similar to previous experiments, we use 800 clients per round and train for 10K rounds with FedAvg. Figure 13 and Table 8 contain the results for the large models with and without the efficient techniques applied. We apply two levels of quantization on download, 16 and 32 bits, and observe that the Large LSTM is more amenable to download quantization compared to the Large Transformer as the regression between the two levels is much smaller for the LSTM than the Transformer. However, the Transformer with 16 bit download quantization still outperforms all efficient LSTMs though it requires more communication rounds to do so than the efficient Transformer with 32 bits for download. For the remaining analysis, we focus on the efficient Transformer using 32 bits for download. It is clear that for the Large Transformer, applying efficient techniques yields better quality in earlier communication rounds. Although there are regressions in the final model quality after 10K rounds of training, this could be attributed to previously observed issues with increased amounts of labeled data diminishing the value pretraining (Zoph et al., 2020). However, the Efficient Large Transformer still reaches the same final perplexity as the Large LSTM which had no efficient techniques applied. Furthermore, when considered in terms of actual communication cost, as is done in Figure 8, the efficient models yield much better performance at smaller total communication costs.

Table 7: Test perplexity after 10K communication rounds of training for each class of model and federated algorithm.

| Model | Algorithm | Perplexity |
|---|---|---|
| Small LSTM | FedAvg | 34.80 |
| Small LSTM | MimeLite | 34.81 |
| Small LSTM | FedProx | 34.66 |
| Small Transformer | FedAvg | 38.66 |
| Small Transformer | MimeLite | 39.88 |
| Small Transformer | FedProx | 38.57 |
| Large LSTM | FedAvg | 30.83 |
| Large LSTM | MimeLite | 31.00 |
| Large LSTM | FedProx | 30.76 |
| Large Transformer | FedAvg | 29.15 |
| Large Transformer | MimeLite | 30.39 |
| Large Transformer | FedProx | 29.04 |

Table 8: Test perplexity and total communication costs in gigabytes after 10K communication rounds of training for each class of model and setup. If the number of download bits is unspecified, the standard 32 bits was used.

| Model | Download Cost (GB) | Upload Cost (GB) | Perplexity |
|---|---|---|---|
| Small LSTM | 188 | 188 | 34.80 |
| Small Transformer | 164 | 164 | 38.66 |
| Large LSTM | 752 | 752 | 30.83 |
| Large Transformer | 840 | 840 | 29.15 |
| Efficient Large LSTM (download 32 bits) | 752 | 75 | 32.57 |
| Efficient Large Transformer (download 32 bits) | 840 | 84 | 30.83 |
| Efficient Large LSTM (download 16 bits) | 376 | 75 | 32.76 |
| Efficient Large Transformer (download 16 bits) | 420 | 84 | 32.32 |

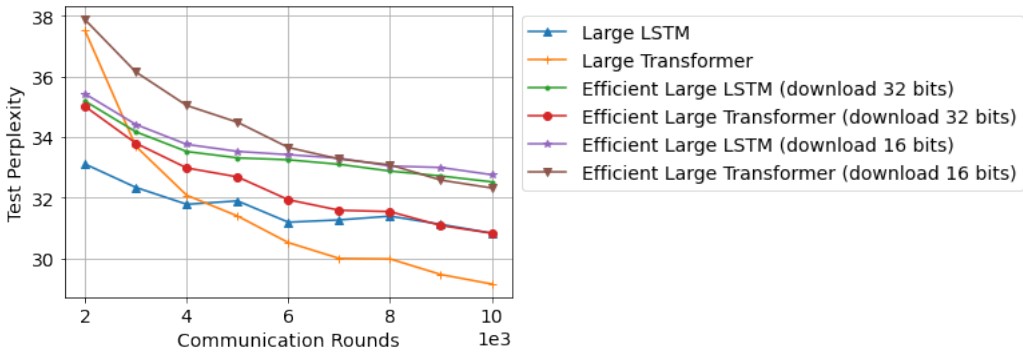

Figure 13: Test perplexity over communication rounds for the large models with and without efficient techniques applied.