# OpenReview forum: "Scaling Language Model Size in Cross-Device Federated Learning"
_aclweb.org/ACL/2022/Workshop/FL4NLP — FL4NLP@ACL2022_

### Official Review · Reviewer_HMNd · 2022-03-24

**Rating:** 8
**Confidence:** 4

**Review:**

**Summary**
The paper studies cross-device federated learning for the problem of language modelling. The paper conducts a series of empirical studies to show that large and high-performing language models can be trained in the cross-device setting, with large Transformers partially fine-tuned through federated updates and quantized communication achieving very good performance.

**Overall comments**
The paper conducts a careful empirical study on how high-performing language models can be trained in the cross-device setting. While the techniques and methods employed in the paper already exist in the literature, the empirical results demonstrated in the paper has high practical value (and would therefore be considered as significant) to practitioners. The paper is therefore of high quality. The paper is also written clearly. I therefore recommend acceptance.

---

### Official Review · Reviewer_D9ht · 2022-03-25

**Rating:** 9
**Confidence:** 4

**Review:**

**Summary Of The Paper:** This paper leverages several techniques for mitigating the communication and computation bottlenecks to train a Transformer in cross-device federated learning. They systematically evaluate partial model training, quantization, efficient transfer learning, and communication-efficient optimizers,

**Strengths.**
- This paper is easy to read and has good structure, and is complete and coherent.
- The topic is interesting and important because the large transformer model has become mainstream in the NLP community. It is necessary to consider how to deploy this kind of large model in the client.
- The design of experiments is sufficient and comprehensive.

**Weaknesses.**
- All the methods are general so it would be better to design a novel method. However, I think it is okay for an analytical paper.

Overall, in my view, this is a high-quality paper.

---

### Official Review · Reviewer_kZQV · 2022-03-26
**Overall nice work analyzing different ways to scale federated training of transformers**

**Rating:** 7
**Confidence:** 3

**Review:**

## Summary
This paper primarily studies the effect of partial variable training, quantization, and their combinations to enable training large language models in a federated setup. These techniques allow training large language models in cross-device federated configurations. They show that quantizing the models before uploading and downloading can reduce training costs with marginal drops in performance.

### Main Observations:
- Model quantization before uploading from local learners is more effective than quantizing before downloading to the local learner.
- Partial variable training combined with quantization further reduces the training and communication load.
- Transfer learning or pretraining can speed up federated training.

## Suggestions:
Overall, I like the paper's analysis of different options available for scaling up federated training for language models and the paper is well-written. However, it still lacks comparison with prior works and baselines. For example, an alternative approach to PVT could be model pruning. So, it would be nice to compare with model pruning baselines such as TPrune ([https://dl.acm.org/doi/10.1145/3446640](https://dl.acm.org/doi/10.1145/3446640)).

Also, please clarify the following in writing:
- How was tokenization done for federated experiments --- centralized or federated? Please clarify.
- Do you exchange model parameters or the changes in parameters for upload and download? While this may be a minor detail, it could affect quantization performance.

---

### Decision · Program_Chairs · 2022-03-26

Accept